# Biosorption of Zn(II) from Seawater Solution by the Microalgal Biomass of *Tetraselmis marina* AC16-MESO

**DOI:** 10.3390/ijms222312799

**Published:** 2021-11-26

**Authors:** Ronald Huarachi-Olivera, María Teresa Mata, Jorge Valdés, Carlos Riquelme

**Affiliations:** 1Centro de Bioinnovación Antofagasta (CBIA), Facultad de Ciencias del Mar y de Recursos Biológicos, Universidad de Antofagasta, Antofagasta 1270300, Chile; maria.mata@uantof.cl (M.T.M.); carlos.riquelme@uantof.cl (C.R.); 2Programa de Doctorado en Ciencias Biológicas, Mención en Biología Celular y Molecular, Facultad de Ciencias de la Salud, Universidad de Antofagasta, Antofagasta 1270300, Chile; 3Laboratorio de Sedimentología y Paleoambientes, Instituto de Ciencias Naturales A. Von Humboldt, Facultad de Ciencias del Mar y de Recursos Biológicos, Universidad de Antofagasta, P.O. Box 170, Antofagasta 1270300, Chile; jorge.valdes.saavedra@uantof.cl

**Keywords:** autofluorescence, adsorption, microalgal, Langmuir

## Abstract

Biosorption refers to a physicochemical process where substances are removed from the solution by a biological material (live or dead) via adsorption processes governed by mechanisms such as surface complexation, ion exchange, and precipitation. This study aimed to evaluate the adsorption of Zn^2+^ in seawater using the microalgal biomass of *Tetraselmis marina* AC16-MESO “in vivo” and “not alive” at different concentrations of Zn^2+^ (0, 5, 10, and 20 mg L^−1^) at 72 h. Analysis was carried out by using the Langmuir isotherms and by evaluating the autofluorescence from microalgae. The maximum adsorption of Zn^2+^ by the Langmuir model using the Q_max_ parameter in the living microalgal biomass (Q_max_ = 0.03051 mg g^−1^) was more significant than the non-living microalgal biomass of *T. marine* AC16-MESO (Q_max_ = 0.02297 mg g^−1^). Furthermore, a decrease in fluorescence was detected in cells from *T. marina* AC16-MESO, in the following order: Zn^2+^ (0 < 20 < 5 < 10) mg L^−1^. Zn^2+^ was adsorbed quickly by living cells from *T. marine* AC16-MESO compared to the non-living microalgal biomass, with a decrease in photosystem II activities from 0 to 20 mg L^−1^ Zn^2+^ in living cells.

## 1. Introduction

Microalgae have a broad spectrum of mechanisms (extracellular and intracellular) to cope with heavy metal toxicity [1]. Microalgae strains revealed varied tolerance and response along with bioaccumulation capability toward heavy metals. Different functional groups, as well as proteins and peptides, are responsible for the metal-binding characteristics. Mechanisms such as extracellular adsorption, reduction, volatilization, complex formation, ion exchange, intracellular accumulation, chelation, and bio-methylation are involved in the bioremediation and biosorption of heavy metals [2]. “Heavy metals” is a term generally given to metals and metalloids with a density of more than 5 g/cm^3^, which include arsenic (As), cadmium (Cd), chromium (Cr), copper (Cu), iron (Fe), lead (Pb), mercury (Hg), silver (Ag), zinc (Zn), and others [3]. Microalgae consume trace amounts of heavy metals such as boron (B), cobalt (Co), copper (Cu), iron (Fe), molybdenum (Mo), manganese (Mn), and zinc (Zn) for the enzymatic processes and cell metabolism. However, other heavy metals such as As, Cd, Cr, Pb, and Hg are highly toxic to microalgae. Due to the hormesis phenomenon, low concentrations of toxic heavy metals can stimulate the growth and metabolism of microalgae [4]. Zinc is a masculine element that balances copper in the body and is also essential for male reproductive activity [5]. Chemically, Zn^2+^ is a Lewis acid that can bind to both the phosphate backbone and the nucleobases of the DNA. Zn^2+^ undergoes hydrolysis even at neutral pH. Furthermore, the partially hydrolyzed polynuclear complexes can affect the interactions with DNA. These features make Zn^2+^ a unique cofactor for DNAzyme reactions [6]. The largest distribution of Zn is in the water of the intertidal zone of the coastal sector of Bahía San Jorge, near the city of Antofagasta, which represents the highest population density in northern Chile. The coastal area of this bay is used extensively as a dumping ground for different sources of pollutants. Zn content varies between 1.7 µg L^−1^ in the northern part of the study area to 2.0 µg L^−1^ in front of the industrial discharge zone. The spatial distribution results from the different local contributions and coastal currents that dilute the contributions and movement of the water toward the North [7].

Experimental tests on the chemical forms of zinc in seawater revealed the following results: (1) ionic plus labile form, (2) inorganic complexes and colloids as well as weak organic complexes, and (3) impeded fraction in large organic molecules and/or organic colloids [8].

Adsorption of heavy metals on the surface of microalgae is a rapid process. It can happen via different paths, including forming a covalent bond between ionized cell walls with heavy metals, ionic exchange of heavy metal ions with cell wall cation, and binding of heavy metal cations with negatively charged uronic acids microalgae exopolysaccharides. However, the process of heavy metal accumulation inside the cell is much slower. The heavy metals are transported actively across the cell membrane and into the cytoplasm, followed by diffusion and subsequent binding with internal binding sites of proteins and peptides such as glutathione, metal transporter, oxidative stress-reducing agents, and phytochelatins [9,10]. Algae biomass adsorbents used for the adsorptive removal of heavy metal pollutants from wastewater show a promising alternative for a single analyte concerning different empirical isotherm models (i.e., Freundlich, Langmuir, Temkin, Sips, and Redlich-Peterson). Freundlich and Langmuir’s models are the most commonly used isotherm models in many studies [11]. For example, blue algae (Cyanophyta) biomass was an excellent sorbent to treat industrial effluents containing Zn^2+^ ions in single metal systems [12].

Photosynthesis is the most basic physiological process of microalgae. Most of the light energy absorbed by chlorophyll is utilized for photosynthesis. The remaining energy is released in the form of heat and fluorescence. Due to the mutual competition between these three methods for energy, photosynthesis can impact fluorescence emissions. Thus, the chlorophyll fluorescence analysis technology developed in recent years has become a new method for measuring and diagnosing plant living bodies based on the theory of photosynthesis [13].

Furthermore, when contaminants interact with algal cells, the toxic effects of pollutants on algae can be expressed through photosynthesis in the form of a change in chlorophyll fluorescence. Therefore, chlorophyll fluorescence as a probe for biological toxicity analysis has become a new method for comprehensively assessing water pollution [14]. At present, the maximum photochemical quantum yield of PSII (F_V_/F_M_) as a photosynthesis activity fluorescence parameter is widely used in the toxicity analysis of pollutants [15,16]. However, the other chlorophyll fluorescent parameters were not very efficient in evaluating the toxic effects of toxicants on algae [14].

Fluorescence recovery after photobleaching (FRAP) has been used to study protein diffusion since the 1970s [17,18]. FRAP technique is widely used in cell biology to observe the dynamics of biological systems, including the diffusion of membrane components providing information on it. The photosynthetic membrane has been considered an ideal model to study protein mobility, given its naturally fluorescent properties [19]. However, the complexity of the photosynthetic membrane should be considered in FRAP, as the numbers of fluorescent proteins are located densely in the membrane with a close functional association promoting efficient energy transfer [20]. Thylakoid membranes are dynamic systems in which the lateral mobility of proteins and lipids plays a crucial role in physiological processes, including electron transport, regulation of light-harvesting, membrane biogenesis, and turnover and repair of proteins. An optimum density of packing of protein complexes into the membrane allows some fluidity in the membrane combined with a high density of photosynthetic complexes and efficient interaction of reaction centers and light-harvesting complexes [21].

The microalga *Tetraselmis marina* AC16-MESO tolerates and removes high concentrations of metal ions. In addition, it is capable of eliminating these metals at a high rate, within a relatively short time, and with a high sedimentation efficiency. These characteristics make *T. marina* AC16-MESO a promising candidate for use in bioremediation [22]. This study aimed to evaluate the adsorption of Zn^2+^ in seawater using the microalgal biomass of *Tetraselmis marina* AC16-MESO “in vivo” and “not alive” at different concentrations of Zn^2+^ (0, 5, 10, and 20 mg L^−1^) at 72 h. In the present research, *T. marina* AC16 MESO “living” and “non-living” microalgal biomass was used as Zn^2+^ bio adsorbent material and subjected to the Langmuir adsorption isotherm model in the seawater solution at different concentrations of Zn^2+^.

## 2. Results

### 2.1. Algal Growth and Maximum Quantum Yield (F_V_/F_M_) from Microalgae Tetraselmis marina AC16-MESO in Bacteriological Agar Medium under Different Concentrations of Zn^2+^

In Figure 1C, the culture of the microalgae *Tetraselmis marina* AC16-MESO in concentrations from 0 to 10 mg/L of Zn^2+^ revealed a constant growth of *T. marina* AC16-MESO, except R3 of concentration 2 and 10 mg/L Zn^2+^. The lowest growth was observed at 20 mg/L Zn^2+^ in the repetitions (R2 and R3).

In Figure 1D, it is observed that the changes in Chl-a fluorescence measured by the optimal quantum yield of photosystem II (PSII, Fv/F_M_), at concentrations of 0.5, 2, 5, 10, and 20 mg L^−1^ Zn^2+^ where the statistical significance between groups was *** *p* < 0.001, ** * p* < 0.01, * *p* < 0.05 with respect to 0 mg/L Zn^2+^ (Tukey’s multiple comparison test).

### 2.2. Zn^2+^ Adsorption Experiments in Living and Non-Living Biomass from Tetraselmis marina AC16-MESO

The fittings of experimental equilibrium to the Langmuir model are represented in Figure 2, and the corresponding fitted parameters are depicted in Table 1. The adsorption capacity of Zn^2+^ of the live microalgal biomass was more significant than the microalgal dead biomass of *T. marina* AC16-MESO, which can be confirmed by the values of Q_max_ in Table 1. According to the Langmuir isotherm (1918) model, the Q_max_, which is the maximum adsorption capacity, corresponds to the saturation of a monolayer of adsorbate molecules on the adsorbent surface. When molecules of adsorbate occupy all adsorption sites, each adsorbent has a unique Q_max_ for each adsorbate. Therefore, Q_max_ is used to predict adsorbent performance in real systems and for the design of adsorbents at different scales [23]. In this work, the Q_max_ determined by the microalgal live biomass (0.03051 mg g^−1^) was significantly higher than the microalgal dead biomass of *T. marina* AC16 MESO (0.02297 mg g^−1^) (Table 1).

### 2.3. Microalgal Growth, Evaluation on the Formation of Chlorophylls “a” and “b”, and Maximum Quantum Yield (Fv/F_M_) under Different Concentrations of Zn^2+^

The effect of Zn^2+^ on *Tetraselmis marina* AC16-MESO is shown in Figure 3A. A higher concentration of Zn^2+^ decreases the microalgal growth determined by counting every 24 h for 3 days.

In Figure 3B,C, the maximum values were found in the negative control (0 mg L^−1^ Zn^2+^). An increase or decrease is observed in both chlorophylls “a” and “b” at different concentrations of Zn^2+^ without significant differences (*p* > 0.05) (Tukey’s multiple comparison test), revealing an intermediate accumulation in concentration of 10 mg L^−1^ Zn^2+^. The lowest concentrations of chlorophylls “a” and “b” are observed in 5 and 20 mg L^−1^ Zn^2+^.

In Figure 3D,E, it can be seen that F_V_/F_M_ is in the typical range of healthy cells (0 mg L^−1^ Zn^2+^: 0.657; 5 mg L^−1^ Zn^2+^: 0.663; 10 mg L^−1^ Zn^2+^: 0.642; 20 mg L^−1^ Zn^2+^: 0.657). After 72 h in Figure 3E, it is observed, under different concentrations of Zn^2+^, in a static system, a decrease in F_V_/F_M_ values is shown below 0.4 where the highest value is observed in control (0 mg L^−1^ Zn^2+^) with a decrease in F_V_/F_M_ from 5, 10, and 20 mg L^−1^ Zn^2+^ without significant differences (*p* > 0.05) (Tukey’s multiple comparison test.

### 2.4. “In Vivo” Fluorescence Analysis for Microalgae Tetraselmis marina AC16-MESO under Different Concentrations of Zn^2+^

Histograms of flow cytometry in autofluorescence of *Tetraselmis marina* AC16-MESO cells are shown in Figure 4. The cell count is along the *Y*-axis and light scattered forward is the *X*-axis (FSC), which is proportional to the size or surface of the cells or particles. The *Tetraselmis marina* AC16-MESO cells exposed to different concentrations of Zn^2+^ showed a decrease in fluorescence. This decrease was in the following order: 0 mg L^−1^ Zn^2+^: 83.8% < 20 mg L^−1^ Zn^2+^: 77.06% < 5 mg L^−1^ Zn^2+^: 69.33% < 10 mg L^−1^ Zn^2+^: 64.33% (Figure 4A–D).

According to Papageorgiou et al. [24] FRAP analysis can be used to assess the mobility of photosynthetic complexes by using the fluorescence recovery time. Microalgae are an excellent model for FRAP because many species have a regular and straightforward thylakoid membrane organization. Therefore, in FRAP analyses at different concentrations of Zn^2+^, an alteration is seen in the fluorescence recovery time, where the shorter recovery time is observed in control (0 mg L^−1^ Zn^2+^). At the same time, the longest recovery time is observed in 20 mg L^−1^ Zn^2+^ and the intermediate times in 5 and 10 mg L^−1^ Zn^2+^ (Figure 5).

## 3. Discussion

In the optimal quantum yield analysis of photosystem II (PSII, F_V_/F_M_) in *T. marina* AC16 MESO cultures, we saw a decrease in F_V_/F_M_ from 0 to 20 mg L^−1^ Zn^2+^. This indicates that the chlorophyll fluorescence parameter F_V_/F_M_ is dose-dependent and time-dependent with Zn^2+^. Furthermore, the photosynthesis activity of *T. marina* AC16-MESO was significantly affected by Zn^2+^. Similar results were found in marine microalgae *Nitzschia closterium* at different concentrations of Pb [14].

Zn^2+^ adsorption capacity was more significant in living biomass than dead biomass of *T. marina* AC16 MESO microalgae. These findings indicate a higher adsorption capacity of Zn^2+^ which is in agreement with previous reports conducted on the Qmax values obtained for the adsorption of Zn^2+^ (1.209 mg g^−1^) on microalgae *Tetraselmis gracilis* (Kylin) Butcher and Zn^2+^ (4.79 mg g^−1^) in blue green (Cyanophyta) living algae [12]. A strain of *Desmodesmus pleiomorphus* was isolated from a strongly contaminated industrial site in Portugal. The metal removal by that strain reached a maximum of 360 mg Zn g^−1^ biomass after 7 days, at 30 mg Zn L^−1^ concentration, in an initial rapid phase of uptake [25]. Suppose the amount of metal added results in solution concentrations exceeding the solubility limit in seawater. In that case, precipitation is predicted to occur, although the kinetics of precipitation are often complex, and the degree of precipitation observed will depend on the duration of the toxicity test. Practically, this causes many problems in understanding the concentration and form of metals that organisms are exposed to in seawater. Both dissolved and particulate metal species may be toxic, and the contributions of these forms to toxicity will likely change with time [26]. The biosorption of the heavy metal Zn^2+^ by dried marine green macroalgae (*Chaetomorpha linum*) was investigated at different solution pH values (2–6), different algal particle sizes (100–800 micron), and different initial metal solution concentrations (0.5–10 mM). The dried alga produced maximum zinc uptake values (Q_max_) of 1.97 mmol g^−1^ (according to the Langmuir model) [27]. The adsorption for Zn (0.243 g g^−1^) was obtained on *Cyclotella cryptica* [28]. The isotherms of Zn sorption by dairy manure at 200 °C were better fitted to the Langmuir model, where the maximum sorption capacity of Zn is 31.6 mg g^−1^ [29]. The accumulation of zinc concentrations was most significant in *Dictyota dichotoma* with a value of 5.117 ± 0.017 µg g^−1^ dry weight. It was relatively uniform in three species of algae, *Ulva lactuca*, *Codium fragile,* and *Jania rubens* at 4.823 ± 0.010, 4.666 ± 0.006, and 4.651 ± 0.017 µg g^−1^ dry weight, respectively [30]. Therefore, live *T. marina* AC16 MESO biomass exhibited the highest value of maximum adsorption capacity compared to non-living biomass.

A higher concentration of Zn^2+^ decreases the growth of *T. marina* AC16-MESO. Similar effects were seen in a study that displayed that nano-ZnO caused a more potent inhibitory impact on microalgal development than bulk-ZnO. Compared with Zn^2+^ released by nano-ZnO into medium, lipid peroxidation injury, aggregation of nano-ZnO, and transmembrane process of nano-ZnO also contributed to toxicity [31]. It has been observed that the red microalga was highly resistant to ZnS nanoparticles, most likely due to the presence of phycoerythrin proteins in the outer membrane-bound Zn^2+^ cations defending their cells from further toxic influence [32]. However, the dissolved Zn^2+^ ions can disturb and inhibit photosynthesis and induce oxidative stress in the algal cell. This resulted in the inhibition of the growth of the microalgae [31], where various microalgal species regulate their cellular content of zinc [33]. The cell density from marine microalgae *Pleurochrysis roscoffensis* after 96 h exposure to five different Zn concentrations (50, 100, 500, 1000, and 1500 µg L^−1^), and a control (free of added Zn), in non-acidified natural seawater revealed significant differences (*p* < 0.01) of microalgae responses between concentrations of 500, 1000, and 1500 µg L^−1^ compared with the algae density of the control [34].

In *T. marina* AC16 MESO, a longer fluorescence recovery time is observed at higher concentrations of Zn^2+^, possibly due to decoupling in the photosystems located in the chloroplast thylakoidal membrane. In microalgae, large chloroplasts consist of sheets arranged parallel without stacking and occupy most of the intracellular space in vivo. Thus, this organization’s regular thylakoid is suitable for FRAP measurements [19,35]. As for the thylakoid membrane dynamics, it may be necessary to add fluorescent labels to membrane proteins that are not naturally fluorescent. However, fusions of the green fluorescent protein (GFP) genes do not appear to work in *Synechococcus* 7942. Yet, GFP can be successfully expressed in other cyanobacteria. Furthermore, many mutant forms are available whose excitation and fluorescence emission do not overlap severely with photosynthetic pigments [36]. In further investigations, direct observations of the fluorescence dynamics of phycoerythrins using FRAP demonstrated energetic decoupling of phycoerythrins in Phycobilisomes (PBsomes) against strong excitation light in vivo, which is projected as a photoprotective mechanism in red algae attributed by the PBsomes in response to excess light energy [37]. The complexity of cellular microenvironments and biological diffusion is often correlated over time and described by a time-dependent diffusion coefficient, D(t). Many efforts have been made to quantify D(t) by FRAP. However, straightforward approaches to quantify a general form of D(t) are still lacking [38].

The living and non-living biomass microalgae *Tetraselmis marina* AC16-MESO was used as a Zn^2+^ bioadsorbent. Regarding the experimental equilibrium results, the Langmuir isotherm model defines the results using the Q_max_ parameter where the live biomass microalgal (Q_max_ = 0.03051 mg g^−1^) value was more significant than the *T. marina* AC16-MESO microalgal dead biomass value (Q_max_ = 0.02297 mg g^−1^). Therefore, it is concluded that the microalgal *T. marina* AC16-MESO biomass can be used as a low-cost biosorbent to remove Zn^2+^ from industrial effluents. Furthermore, in additional investigations conducted on bioadsorption, changes in the fluorescence of microalgae with a decrease in photosystem II activities were observed in the range of Zn from 0 to 20 mg L^−1^.

## 4. Materials and Methods

### 4.1. Microalgae Culture

*Tetraselmis marina* AC16-MESO strain was isolated from the intertidal area of the Bay of San Jorge, Antofagasta, Chile, then identified and preserved in the Marine Mesocosmos Laboratory of the University of Antofagasta, Chile. The strain was cultivated at 20 °C in Erlenmeyer flasks of 250 mL using sterilized seawater with the addition of UMA5 medium [39] (NaNO_3_ 4.55 × 10^−5^ M; NaH_2_PO_4_·H_2_O 2.41 × 10^−4^ M; NaHCO_3_ 1.99 × 10^−3^ M) at 20 °C and with a continuous photosynthetic photon flux of 70 µmol m^−2^s^−1^ (24 h light). The algal cells were counted using a hemocytometer-type chamber in an optic microscope [40,41].

### 4.2. Evaluation of the PSII of Tetraselmis marina AC16-MESO in a Solid Medium in the Presence of Zn^2+^

The PSII of *Tetraselmis marina* AC16-MESO was assessed in f/2 solid medium (NaNO_3_, 75 g L^−1^; NaH_2_PO_4_•2H_2_O 5.65 g L^−1^) [42] containing OXOID marks bacteriological agar at different concentrations of Zn^2+^ (0, 0.5, 2, 5, 10, and 20 mg L^−1^ Zn^2+^). Filtered and autoclaved seawater was used to prepare the medium. The pH was adjusted to 8 with 1 M NaOH and HCl, and 15 g of bacteriological agar per liter was added and autoclaved for 15 min. For the sowing of the microalgae, *Tetraselmis marine* AC16 MESO was added into the Drigalsky glass handle along with 50 µL of liquid culture sample and then emptied into the Petri dishes.

### 4.3. Preparation of Living and Non-Living Microalgal Adsorbent Biomass

The live microalgal biomass of *Tetraselmis marine* AC16 MESO was used for bioadsorption studies to assess the ability to adsorb at different concentrations (0, 5, 10, 20 mg L^−1^ Zn^2+^). These concentrations are in line with those reported by others in the literature [43,44]. The samples were inoculated at 100 mL of logarithmic phase into 250 mL Erlenmeyer bottles. Microalgae were added to flasks containing 100 mL UMA5 medium prepared with sterilized seawater, without trace elements [45]. They were then incubated at 28 ± 2 °C and 70 mmol m^−2^s^−1^, with cold light intensity provided by the white fluorescent lamps for 72 h.

In non-living microalgal biomass assays, the dry biomass used in the adsorption processes was obtained from 2.5 L microalgal culture of non-living *Tetraselmis marina* AC16 MESO with 2,945,000 cell m L^−1^ equivalent wet weight of 45.6 g. The biomass obtained was washed twice with distilled water. Next, it was frozen and lyophilized [46]. Before using as an adsorbent, the lyophilized biomass was ground and homogenized to obtain a “non-living” microalgal biomass wherein each experimental unit contains 0.5 g of dry biomass in a volume of 40 mL of sterile seawater at 0.5, 10, and 20 mg L^−1^ of Zn^2+^ with a test time of 72 h.

### 4.4. Adsorbent Biomass and Adsorption Experiments

To measure Zn^2+^ in living and non-living biomass of the *Tetraselmis marina* AC16-MESO microalgae, 0.5 and 1 g of microalgal biomass sample (in triplicate) was homogenized using an agate mortar in a reflux system with a glass funnel covered with a watch glass. The pieces were placed in a 125 mL beaker for disintegration with 10 mL of HNO_3_ Suprapur and heated on a heating plate at 150 °C for 2 h. Subsequently, the resulting solution was filtered (0.45 µm) and packed into a 25 mL volumetric flask with deionized water. The quantification of metals was performed by atomic absorption spectrophotometry (Shimadzu, AA 6300) by using the flame technique at a wavelength of 213.9 nm with a mixture of C_2_H_2_ gas. The concentrations were expressed as mg g^−1^ wet weight [47].

### 4.5. Langmuir Equation

The balance of the adsorption isotherm of Zn^2+^ was calculated using the Langmuir sorption model [48] with the Origin6 program to determine the maximum adsorption of Zn^2+^ in “living” and “non-living” cells of *Tetraselmis marina* AC16-MESO. The adsorption of Zn^2+^ (q) for the construction of adsorption isotherms was determined using the following equation:q = q_max_bC_f_/(1 + bC_f_)(1)
where q is the adsorption of Zn^2+^ (mg Zn^2+^ g^−1^ dry biomass), q_max_ is the maximum adsorption (mg Zn^2+^g^−1^ dry biomass), Cf is the final concentration of Zn^2+^ in the solution (mg L^−1^), b (L mg^−1^) is the Langmuir constant. Here, adsorption rate is equal to the desorption ratio. For the preparation of experimental data, the Langmuir model was linearized using the following equation:C_f_/q = (1/q_max_b) + (C_f_/q_max_)(2)

### 4.6. Determination of Chlorophylls “a” and “b”

For the quantification of chlorophyll, 2 mL of *Tetraselmis marina* AC16-MESO cell culture were taken for centrifugation and alteration in a sonicator (ultrasonic bath) for 7 min (20 kHz) by extracting with 2 mL of acetone. The extract was centrifuged at 3000× *g* for 5 min, and the optical density was measured at 647 and 664 nm [49]. The concentrations of chlorophyll a and chlorophyll b (µg mL^−1^) can be calculated based on optical density values as follows:Chl a = −1.93 × OD_647_ + 11.93 × OD_664_
Chl b = 20.63 × OD_647_ − 5.5 × OD_664_

### 4.7. Measurement of the Fluorescence of Chlorophyll a (Chl-a) in the Living Culture of Tetraselmis marina AC16-MESO

The maximum quantum yield of PSII (F_V_/F_M_) of *Tetraselmis marina* AC16-MESO exposed to 0, 5, 10, and 20 mg L^−1^ Zn^2+^ was measured in triplicate after 10 min of adapting to darkness by using pulse amplitude modulated chlorophyll fluorometer (JUNIOR-PAM WALZ, Germany). The maximum quantum yield of photosystem II, F_V_/F_M_, was calculated according to Schreiber [50]:F_V_/F_M_ = (F_M_ − F_0_)/F_M_(3)

### 4.8. Flow Cytometric Analysis

The microalgal cells from *Tetraselmis marina* AC16-MESO at different concentrations of Zn^2+^ were suspended in 500 µL of the same medium with 3% formalin, and the autofluorescence emitted due to chlorophyll was observed. The size and complexity of the cells were determined using a BD FACSVerse™ flow cytometer equipped with an air-cooled 488 nm argon-ion laser exciting the cells at that wavelength.

### 4.9. Fluorescence Recovery after Photobleaching (FRAP) Analysis in Microalgal Autofluorescence from Tetraselmis marina AC16-MESO

In the FRAP analysis, live *Tetraselmis marina* AC16-MESO cells immobilized on a slide covered with a glass slip were observed under the objective lens of confocal spectral microscopy at different concentrations of Zn^2+^ (0, 5, 10, and 20 mg L^−1^). Due to the autofluorescence of the microalgae, images were obtained using a TCS SP8 Confocal Spectral microscopy system, mARK Leica, with a 40X objective. Photobleaching was performed in an area marked ROI in the microalgal cell. ROI areas were bleached with 100% Argon-405 power laser, and the images were grouped 2 × 2 to increase signal noise per camera.

After opening the laser shutter for 448 milliseconds (ms), images of the cells were taken by microscopy every 0.148 s for pre-whitening, bleaching, and post-whitening with a total duration of 448 ms and a total exposure time of 32.55 s. Then, the results were exported to Excel. The fluorescence intensity was calculated as MFI. The final data were analyzed using Graph Pad Prism 6.0.

### 4.10. Data Analysis and Statistics

For the study, the alteration of FRAP in *Tetraselmis marina* AC16-MESO at different concentrations (0, 5, 10, and 20 mg L^−1^) of Zn^2+^ were evaluated using the cell growth, maximum photochemical efficiency, fluorescence, and fluorescence recovery after photobleaching (FRAP) analysis. The mean and standard error were calculated to obtain a representative “average” at the end of 72 h. The ANOVA was performed with three repetitions in each treatment. The results were considered significant at *p* < 0.05.

## Figures and Tables

**Figure 1 ijms-22-12799-f001:**
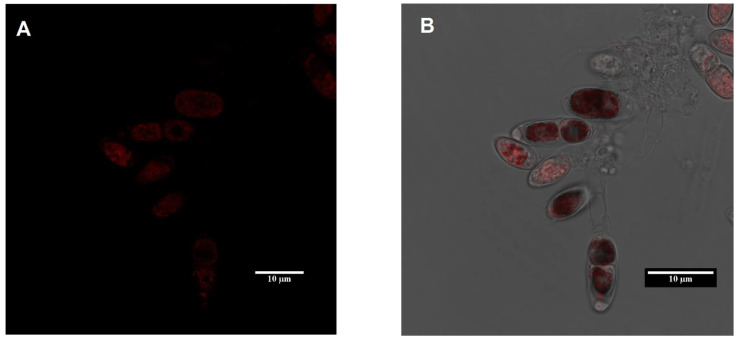
Observation of the microalgae *Tetraselmis marina* AC16-MESO. (**A**,**B**) Photographs of microalgal cells through confocal microscopy. (**C**) Petri dishes with bacteriological agar medium at different concentrations of Zn^2+^ after 68 days. (**D**) Maximum quantum yield (F_V_/F_M_) in cultures of *T. marina* AC16 MESO exposed to different Zn^2+^ treatments for 68 days. Bars indicate the standard error (SE), *** *p* < 0.001, ** *p* < 0.01, * *p* < 0.05 with respect to 0 mg/L Zn^2+^ (Tukey’s multiple comparison test).

**Figure 2 ijms-22-12799-f002:**
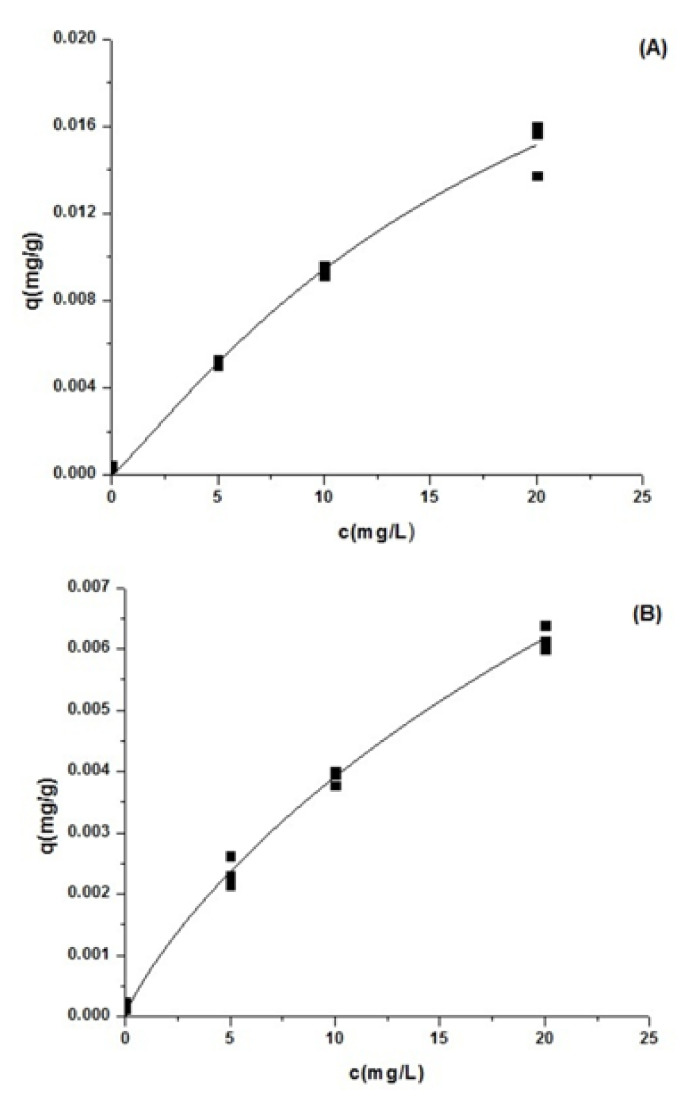
Equilibrium results of the adsorption of Zn^2+^. (**A**) *Tetraselmis marina* AC16-MESO living biomass; (**B**) *Tetraselmis marina* AC16-MESO non-living biomass. Experimental data on the equilibrium adsorbed concentration of Zn^2+^ (q, mg·g^−1^) versus the equilibrium Zn^2+^ concentration in the liquid phase (C, mg·L^−1^) represented together and fit into the Langmuir equilibrium isotherm models.

**Figure 3 ijms-22-12799-f003:**
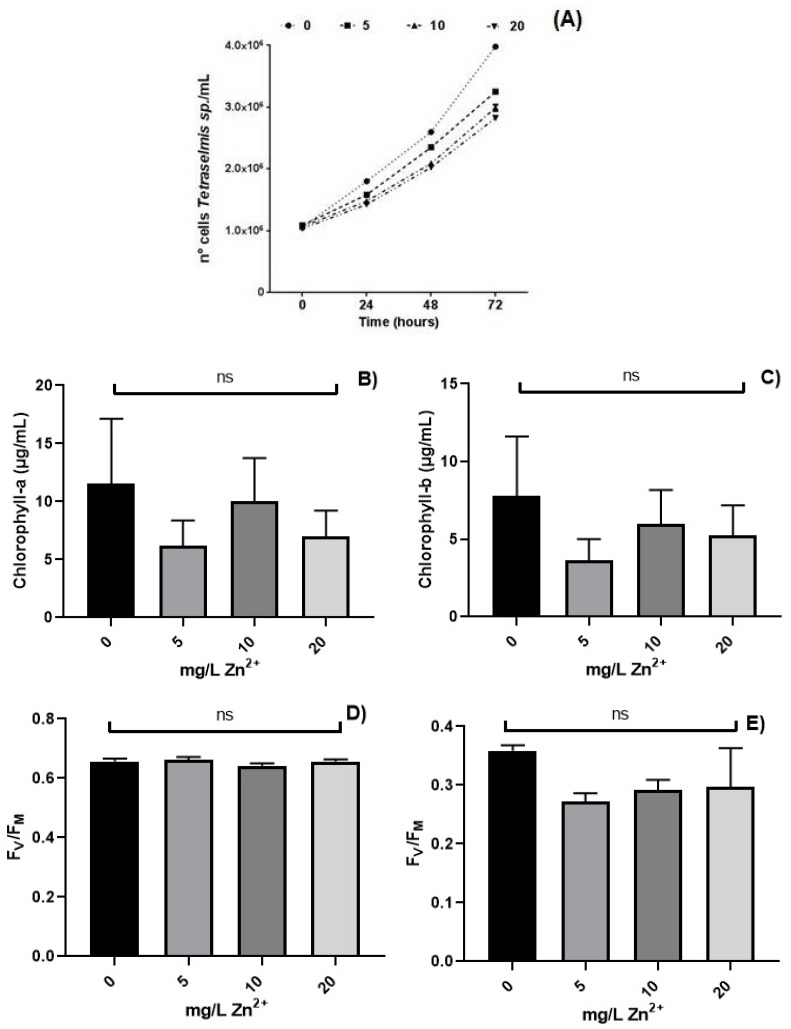
Effect of concentration Zn^2+^(mg/L): (**A**) growth microalgal *Tetraselmis marina* AC16-MESO at 24, 48, and 72 h. Concentrations of the photosynthetic pigment at 72 h: (**B**) chlorophyll “a”; (**C**) chlorophyll “b”. Maximum quantum yield (F_V_/F_M_) in *T. marina* AC16-MESO at (**D**) 0 h. (**E**) 72 h of exposure. Bars indicate the standard error (SE); ns indicates not significant, (*p* > 0.05). 5 vs 0; 10 vs 0; 20 vs 0 (mg/L Zn^2+^).

**Figure 4 ijms-22-12799-f004:**
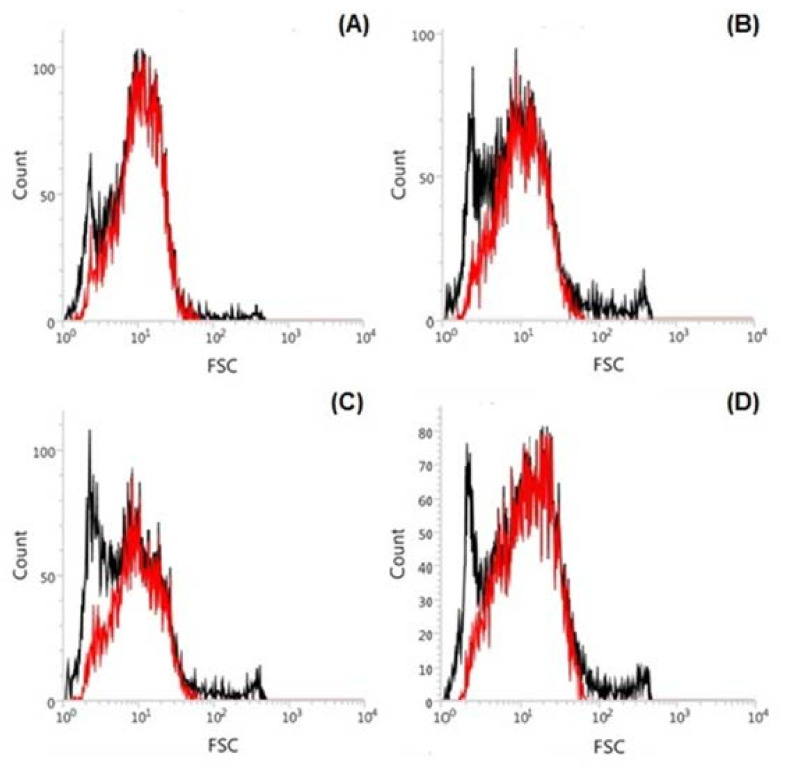
Graphical representation of the autofluorescence signal movement in *Tetraselmis marina* AC16 MESO (red curves). Flow cytometer histograms representing the excited autofluorescence signal in cells of *T. marine* AC16 MESO with 488 nm blue laser, at different concentrations of Zn^2+^. (**A**) 0 mg/L Zn^2+^; (**B**) 5 mg/L Zn^2+^; (**C**) 10 mg/L Zn^2+^; (**D**) 20 mg/L Zn^2+^.

**Figure 5 ijms-22-12799-f005:**
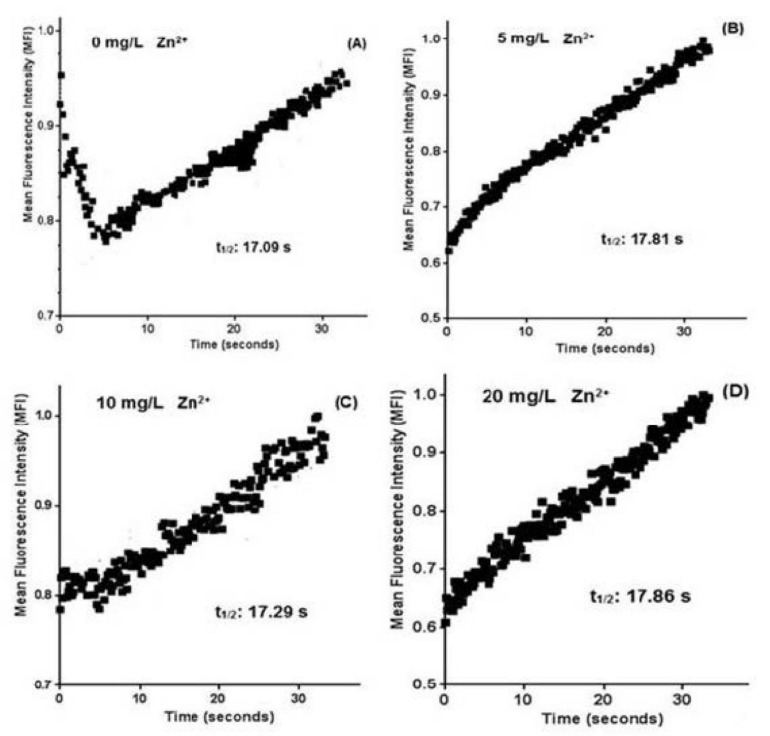
Analysis of fluorescence recovery after photobleaching (FRAP) in autofluorescence of native pigments in microalgae *Tetraselmis marina* AC16-MESO, at different concentrations of Zn^2+^ after 72 h of exposure. (**A**) 0 mg L^−1^; (**B**) 5 mg L^−1^; (**C**) 10 mg L^−1^; (**D**) 20 mg L^−1^.

**Table 1 ijms-22-12799-t001:** Parameters from fitting the experimental results with equilibrium isotherm (Langmuir equilibrium isotherms) model.

Equilibrium Isotherms			
Model	Parameter	Biomass Microalgae *Tetraselmis marina* AC-16 MESO	
		Non-Living(mg/g)	Living (mg/g)
	Q_max_	0.02297	0.03051
Langmuir	K	0.02964	0.03249
	n	0.84065	1.13885
	r^2^	0.99345	0.98789

## Data Availability

Data available upon request.

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
