# Peer review of "Biosorption of Zn(II) from Seawater Solution by the Microalgal Biomass of Tetraselmis marina AC16-MESO"

_ijms, 2021, doi:10.3390/ijms222312799_

Round 1
Reviewer 1 Report
Manuscript is valuable and well prepared but some improvements are necessary. Firstly, add aim of research in the end of introduction. Was there a correlation between used doses of Zn and observed in the environment real values of Zn concentration? Please explain why those doses were used in research. Adsorbent biomass and adsorption experiments were in fact conducted in non living biomass in both situations. After heating in 150oC There were not living cells. Do you agree? Statistical methods were used. It should be used in the results description. Please add information’s when differences were significant.
Author Response
RESOLUTION FOR THE FIRST REVIEWER
First question: The manuscript is valuable and well prepared, but some improvements are needed. First, add the research objective at the end of the introduction. Was there a correlation between the used doses of Zn+2 and the actual values ​​of Zn+2 concentration observed in the environment? Explain why those doses were used in the research.
Answer: The concentrations of 0, 5, 10 and 20 mg/L of Zn2+ were taken based on the evaluations carried out in the following investigations, mentioning in the manuscript in lines 359 and 360 highlighted in yellow:
1.- Zhou, G.J., Peng, F.Q., Zhang, L. J., Ying, G.G. (2012). Biosorption of zinc and copper from aqueous solutions by two freshwater green microalgae Chlorella pyrenoidosa and Scenedesmus obliquus. Environmental Science and Pollution Research, 19 (7), 2918-2929. https://doi.org/10.1007/s11356-012-0800-9
2.- Alam, M.A., Wan, C., Zhao, X.Q., Chen, L.J., Chang, J.S., Bai, F.W. (2015). Enhanced removal of Zn2+ or Cd2+ by the flocculating Chlorella vulgaris JSC-7. Journal of Hazardous Materials, 289, 38-45. https://doi.org/10.1016/j.jhazmat.2015.02.012
Was added two bibliographic sources shown in lines 575 to 579
Second Question: In fact, the adsorbent biomass and adsorption experiments were carried out on non-living biomass in both situations. After heating to 150 ° C, no living cells remained. Do you agree? Statistical methods were used. It should be used in the description of the results. Add information when the differences are significant.
Answer: It is NOT correct that non-living biomass was used in both situations, because during the bioadsorption process with living biomass, living cells were used evaluating the bioadsorption of Zn2+, maximum photochemical efficiency, cell density, fluorescence by flow cytometry as well. as FRAP in confocal fluorescence microscopy. Heating to 150 °C was carried out after the experimentation process for the measurement of Zn2+. On the other hand, during the bioadsorption process with "non-living" biomass, lyophilized biomass was used, selecting the 45-day cultures with the highest microalgal growth, free of contamination, using 0.5 g of dry biomass in a volume of 40 mL of medium at different concentrations of Zn2+. during the experimentation process, which was corrected in line 366, highlighted in yellow.
Too the statistical term was added "with significant differences (P < 0.05). " in line 212 and 213
The objective was added to the end of the introduction shown in the lines 124 to 127
Reviewer 2 Report
The aim of this research was to study the use of T. marina AC16 MESO “living” and “non-living” microalgal biomass as Zn2+ bio adsorbent material and subjected to the Langmuir adsorption isotherm model in the seawater solution at different concentrations of Zn2+. It is an interesting topic and authors managed to provide a clear statement of the scientific area, a range of research on the topic, critically analyse the selected topic using published data and provide an indication of what further research is necessary. However, authors could try to organize in paragraphs the presentation of the Results section. In this view, it is suggested the acceptance of the manuscript for publication in IJMS after minor revision.

Author Response
RESOLUTION OF THE SECOND REVIEWER
SHOWN IN ATTACHED PAPER "ijms 1449092 11-11-2021" Where the presentation of the Results section was organized in paragraphs, under the following subtitles
2.1. Algal Growth and Maximum quantum yield (FV/FM) from microalgae Tetraselmis marina AC16-MESO in bacteriological agar medium under different concentrations of Zn2+ on lines 133 to 135
2.2. “Zn2+ adsorption experiments in living and non-living biomass from Tetraselmis marina AC16-MESO” on lines 156 to 157
2.3. “Microalgal growth, evaluation on the formation of chlorophyll-“a” and “b” and Maximum quantum yield(Fv/FM) under different concentrations of Zn2+” on lines 188 to 190
2.4. “In vivo” fluorescence analysis for microalgae Tetraselmis marina AC16-MESO under different concentrations of Zn2+” on lines 214 to 215